# Experience of Hereditary Amyloidosis with Rare Variant in Ecuador: Case Reports

**DOI:** 10.3390/medsci12040058

**Published:** 2024-10-21

**Authors:** Diana Elizabeth Luzuriaga Carpio, Borys Roberto Abrigo Maldonado, Humberto Villacorta

**Affiliations:** 1Hospital General Manuel Ygnacio Monteros-IESS, Loja 110150, Ecuador; 2Heredity S.A., Quito 170135, Ecuador; borys.abrigo@unl.edu.ec; 3Division of Cardiology, Universidad Federal Fluminense, Niterói 24020-141, Río de Janeiro, Brazil; hvillacorta@id.uff.br

**Keywords:** amyloidosis, cardiomyopathy, transthyretin, polyneuropathy, mutation

## Abstract

More than approximately 120 transthyretin mutations are known. Their clinical presentation is heterogeneous, as the course of disease onset depends on genetic variation and level of penetrance. They are little known in Ecuador, and some of the reported cases suggest—given analysis of family trees—that they come from a province that is possibly considered endemic. The main objective of this study is to perform a descriptive observational analysis on the presentation of transthyretin amyloidosis in families carrying the p.Ser43Asn gene of the identified index case.

## 1. Introduction

Transthyretin amyloidosis is a rare, multisystem, underdiagnosed disease that has a great impact on quality of life, with increased mortality when there is cardiac involvement. It should be diagnosed in a timely way so that measures can be sought for multidisciplinary follow-up [1]. There are two forms of transthyretin amyloidosis: the wild form and the genetic form. Some mutations are more frequent and others are more rare.

Transthyretin amyloidosis is described as having a heterogeneous natural history, and therefore a poor prognosis, with a mean survival from diagnosis of 3.6 years for wild-type amyloidosis and 2.5 years for transthyretin amyloidosis of the Vall 122lle variant.

In our first case, the time elapsed to establish a diagnosis of transthyretin amyloidosis from the onset of symptoms was 7 years, through the retrospective analysis of the clinical history of a patient with a mixed phenotype treated at Manuel Ygnacio Monteros-IESS Loja General Hospital. This demonstrates the reality of the existing gaps in the health system in Ecuador and the need to carry out studies on the presentation of transthyretin amyloidosis.

The identification of the first case of transthyretin amyloidosis diagnosed at Manuel Ygnacio Monteros Hospital with the pSer43Asn variant has motivated us to carry out a descriptive study of those index cases identified as carriers of the mutant transthyretin gene. The review of this work will allow the establishment of inter-institutional behaviors and protocols for the future management of this rare pathology, one which is not very well known by Ecuadorian society.

The delay in diagnosis and the rapid progression after the appearance of symptoms led to the death of our first patient, with manifestations of hypertrophic cardiomyopathy, heart failure and sensory-motor polyneuropathy [1].

We consider it necessary to make a timely and early diagnosis, to surveil and monitor clinical manifestations and their time of presentation and to support this monitoring by multidisciplinary management in order to generate comprehensive treatment for each patient; this will allow us to avoid systemic damage, fatal outcomes, higher costs due to hospitalizations and more complex therapies that increase hospital costs.

There are no laboratories in Ecuador that perform the molecular study for familial amyloidosis, either in the comprehensive public, complementary or private health networks. Genetic studies are carried out abroad through private intermediary companies that are responsible for processing the delivery of biological materials to international genetic laboratories.

The burden of heart failure disease in Ecuador is continuously increasing. During the 2014–2018 period, 90,242 years of healthy life (DALY) were lost, of which 46.72% are represented by years of life lost due to premature death (YLL), and 53.28% by years lived with disability (YLD) [2]. The prevalence of amyloidosis in Ecuador is unknown, and the question remains as to how many of our deceased heart failure patients had a rare underlying disease such as amyloidosis. It is difficult to carry out a post-mortem study to answer this question; however, monitoring those identified as carriers will allow us to know what the first manifestations of cardiac amyloidosis caused by the p.Ser43Asn variant are, which will allow us to put the country on alert.

The objective of this article is to perform a descriptive analysis on the presentation of transthyretin amyloidosis in families carrying the p.Ser43Asn gene based on the index case identified in the period from January 2020 to January 2022.

## 2. Case Report

### 2.1. Case Report Number One

A 59-year-old male patient, born in the city of Loja, is seen in a private outpatient service on 30 September 2020, presenting with episodes of lipothymia, dyspnea at rest, generalized weakness, limb edema for 1 month, hyporexia and abundant diarrheal stools 4 to 6 times a day—the same condition that was reported to be chronic for more than 4 years and that was initially classified as irritable bowel syndrome; when consulted it was reported to be more frequent in the last 2 years and 6 months, with significant weight loss. Additional pathological history includes implantation of an ICD device in 2018, due to diagnosis of Hypertrophic Cardiomyopathy, and a diagnosis of ALS (Amyotrophic Lateral Sclerosis). He received treatment for intestinal tuberculosis 1 year prior without clinical improvement. Physical examination: pale facies; cachexic; BP: 81/52 mmHg; HR: 80 bpm; SatO2: 90%; cardiac auscultation with normal sounds; lungs: ventilated; lower limbs with pretibial edema +2/4. Usual medication: propranolol 10 mg every 48 h, amiodarone 200 mg daily, furosemide 10 mg daily. The electrocardiogram study revealed sinus rhythm and inferior pseudo Q. An echocardiogram study was performed during the consultation, observing left ventricular hypertrophy (SIV: 19 mm), right ventricular hypertrophy (Figure 1), thickening of the interatrial septum, global hypokinesia of the left ventricle with LVEF: 24%, decreased velocities on tissue Doppler, diastolic dysfunction grade III, global strain around −5% and mild pericardial effusion (Figure 2). Laboratory studies: elevation of NTproBNP: 13,430 pg/mL; troponin T: 186 pg/mL; urea: 60; creatinine: 1.2 mg/dL; and chronic iron deficiency anemia. It was necessary to perform new evaluations, including neurology–neurophysiology, which determined the presence of severe sensory-motor polyneuropathy. A skin biopsy study was requested, which was negative (kappa and lambda light chain studies were negative)., The presence of systemic lupus erythematosus was determined by the hematology service as an additional finding to the suspicion of heredofamilial amyloidosis due to transthyretin, which was the first presumptive diagnosis after the first evaluation. The patient was evaluated in a hospital unit of the Social Security Institute—Manuel Ygnacio Monteros Hospital, where he continued his evaluations and follow-up. Scintigraphy studies with Tc99m pyrophosphate are only available in the city of Quito, and magnetic resonance studies were not carried out due to lack of availability in the sector. A complete sequencing study of the TTR gene was requested, with the result obtained 3 months after taking the sample: a pathogenic heterozygous variant, position: chr18:31,592,954, variation p.Ser43Asn.

The patient was admitted to another hospital unit privately in February 2021 due to an episode of syncope and trauma to the spine and died on 3 August 2021.

When performing a retrospective analysis of his clinical history, initial medical attention was found with gastrointestinal manifestations since 2013: diarrheal stools alternating with constipation, classified as irritable bowel syndrome. His first registered consultation with cardiology was in 2016, due to the presence of dizziness and isolated ventricular extrasystoles, with a delay in diagnosis of approximately 5 years.

#### Family Tree Analysis

Carrying out the family group analysis allowed us to observe a total of 11 siblings of the index patient, making this family group a total of 33 people, of which 19 were tested, with a total of 5 people with a positive result; 11 did not perform the test, and 5 of these died: 1 from SARS-COV2 pneumonia, 1 from acute myocardial infarction (died in the province of Pichincha due to a diagnosis of Hypertrophic Cardiomyopathy plus acute myocardial infarction) and 3 of unknown cause with probable death from acute myocardial infarction (Figure 2). When inquiring about the origin of their parents, they claim to be from the Quilanga canton in the province of Loja. This family group has members residing in the Province of Loja.

### 2.2. Case Report Number Two

A 49-year-old patient from Loja came to the outpatient clinic at Manuel Ygnacio Monteros-IESS Loja Hospital due to significant edema in the lower limbs. He denied dyspnea or precordial pain. Other reported symptoms: abdominal distention, diffuse abdominal pain and nausea. History of heart failure with reduced LVEF, with suspicion of cardiac amyloidosis according to a report from Guayaquil hospital 6 months ago. On physical examination: BP: 90/50 mmHg; HR: 60 bpm; and SatO2: 93%. Rhythmic heart, without murmurs. Lungs: MV preserved. Predominantly cardiac clinical presentation with symptoms of heart and gastrointestinal failure (abdominal distension and diarrhea). Electrocardiogram: First degree AV block, complete right bundle branch block, inferior pseudoinfarction (Figure 3A). Transthoracic echocardiogram: predominantly septal left ventricular hypertrophy (SIV: 24 mm), right ventricular hypertrophy, thickening of the interatrial septum and atrioventricular valves. Global hypokinesia, LVEF: 35%, global longitudinal strain: −8%. Mild pericardial effusion (Figure 3B). Cardiac magnetic resonance study: subendocardial pattern suggesting amyloidosis. Skin and adipose tissue biopsy negative for Congo red stain. Kappa and lambda light chain tests were negative. Gastric and colon biopsy study negative for Congo red stain. The requested genetic study demonstrates heterozygosity, with the p.Ser43Asn variant. The patient was discharged with a heart transplant referral to a level III hospital.

Family Tree Analysis

The analysis of the family tree allows us to observe a total of approximately 68 members, of which 13 have been tested, with a positive result in 3; the remaining 55 are yet to be tested. There are members who died before this research: 4 people from the second generation, with myocardial infarction and heart failure (Figure 4). The origin of the family line is from the Quilanga canton, in the province of Loja. This family group has members residing in the provinces of Loja, Zamora and Guayas and members residing in Spain.

### 2.3. Case Report Number Three

Patient, 62 years old, male, from the province of Loja, resident in Quilanga, transferred from another hospital unit for pacemaker placement due to a diagnosis of third-degree AV block, hypertrophic cardiomyopathy and heart failure, reporting a 5-month clinical condition of general malaise and dyspnea with medium effort that progresses to small effort accompanied by adynamic and generalized edema. He presented 72 h prior with oppressive precordial pain with nausea and dizziness. Vital signs upon admission: BP: 99/79 mmHg, HR: 36 bpm, RR: 16 per min, T: 36.2 °C and pulse rate: 93% with FiO2 at 24%.

The electrocardiogram study showed complete AV block; he was implanted with a pacemaker and evaluated with an echocardiogram study after implantation: LVEF: 24%; significant hypertrophy of the left ventricle with measurement of the interventricular septum around 33 mm; LV diastolic diameter of 26 mm; 28 mm posterior wall’ right ventricular systolic dysfunction; TAPSE: 13 mm; CAF: 25%. Global Longitudinal Strain: −9% (Figure 5).

His neurological evaluation demonstrates the presence of stage III sensory-motor polyneuropathy.

The genetic study result on 7 July 2022 was complete sequencing with a positive result for the p.Ser43Asn variant, with heterozygosity.

The patient died at home on 17 March 2023.

Family Tree Analysis

The family chart graph indicates a total of 51 members (Figure 6). Of a total of 8 siblings of the index patient, 7 have died, with the place of death in the province of Loja. The causes of death have no established diagnosis within the family history; however, close relatives express that the deceased relatives presented symptoms of generalized weakness, cachexia and dyspnea, with an outcome of sudden death. There is a history of consanguinity in the ancestors of the family. The origin of the family from the Quilanga canton, Loja province.

Of the patients tested, one is in the Province of Oro, the rest are in the Malacatos canton, Loja province.

The following table shows the baseline demographic characteristics and the clinical characteristics of ECG and imaging of the 3 clinical cases (Table 1).

The 3 clinical cases share the same reported variant (Table 2).

## 3. Discussion

Transthyretin amyloidosis is a systemic disease of variable incidence which is still unknown in various parts of the world, with an incidence of 1 per 100,000 inhabitants worldwide [3]. There are some variants that are considered more frequent in certain countries, and they are stipulated as endemic variants in Spain (Mallorca), Portugal, Norway, Japan and Brazil. The TTR c.128G > A; p.Ser43Asn variant, also known as Ser23Asn, is a point mutation in exon 2, codon 43 of the TTR gene that leads to the substitution of a single amino acid of the neutral, polar serine with a neutral, polar asparagine.

It is described in the literature in individuals affected by transthyretin-related amyloidosis and reported in ClinVar, Variation ID: 661615, (public file to support accumulating evidence of clinical significance of genetic variants), and is not found in general population databases (Exome Variant Server, Genome Aggregation Database), indicating that it is not a common polymorphism.

Functional analyses demonstrate reduced monomer stability and are positive for amyloid formation on endomyocardial biopsies from multiple. The serine at codon 43 is moderately conserved, and computational analyses (SIFT: tolerated, PolyPhen-2: possibly damaging) predict conflicting effects of this variant on protein structure/function. Based on available information, this variant is considered to likely be pathogenic.

The three clinical cases share the same variant reported in Table 2 and further evidenced in Figure 7 for the amino acid positions (highlighted in green) subjected to the mutations described in the same table.

It is considered a rare variant [4], with little information on its clinical presentation and on the real worldwide prevalence; there are few case reports on this particular variant. Maria Papahanasiou et al. reported two opportunely diagnosed patients from a family in Italy with genetic variants similar to the one identified in our cases; these variants are considered to have a clinically aggressive course, due to their exclusively cardiac phenotype [5].

There are at least five studies, in addition to the one already mentioned above, that are clinical case reports based on the Ser23Asn variant, which corresponds to the same variant—this designation is due to a change in nomenclature. Currently, the amino acid sequence from 1 to 20 is the signal peptide and the final protein structure is expressed from amino acids 21 to 147, varying the numbering but corresponding to the same amino acid. The variant causes hereditary transthyretin amyloidosis, and this protein has a molecular weight of 14 kDa in monomeric form and exists in tetrameric form (55 kDa) in plasma. As mentioned, it is synthesized in the liver and in the choroid plexus as a homotetrameric structure with a dimer of dimers of quaternary structure [6,7,8].

In 1999, Lawreen Helter Connors et al. mention the case of a 41-year-old patient from northern Portugal as the first study published with this variant, with a manifestation of heart block that required a pacemaker and heart transplant, as is the case with Daoko et al., in their publication in 2010 of a patient from Peru who had heart failure.

In another review of clinical cases published by Colombian authors, they highlight the relevance of the use of the Technetium 99 pyrophosphate scintigraphy technique for the diagnosis of hereditary transthyretinal amyloidosis in a 41-year-old Ecuadorian patient with symptoms of fatigue, difficulty breathing and peripheral neuropathy, giving the particularity of mixed phenotypic expression, whose identified variant corresponds to Ser23ASn, that possibly has Spanish origin. Different researchers mention a possible origin of the variant in Portugal, Spain and Italy [9]; it is difficult to precisely identify the origin of the variant and its presence in the equator is attributed to migration and colonization processes, with a possible founder effect in the province of Loja, particularly in the Quilanga canton.

In Ecuador, some clinical cases of amyloidosis have been documented since 2018 without specifying the genetic type. This is explained by the limitations of the public health system in accessing imaging studies such as cardiac scintigraphy and cardiac magnetic resonance, which are very scarce and constitute a high cost for patients, with limited access for the public sector.

In the city of Loja, capital of the province of Loja in the South of the country, our first case of Transthyretin amyloidosis was diagnosed in 2020 with genetic determination of the variant. The patient presented with hypertrophic cardiomyopathy and sensory-motor polyneuropathy; the main manifestations of amyloidosis that led to death. In the retrospective analysis of the first symptoms, gastrointestinal discomfort was found as the first manifestation: diarrhea and constipation together with the presence of arterial hypotension and dizziness, which confers a mixed phenotype presentation with a delay time of 7 years.

The second case shows gastrointestinal discomfort as first manifestation. There is no neurological involvement, and this case was considered to have an exclusively cardiac phenotype. In the third case, the mixed phenotypic presentation is repeated, initial gastrointestinal manifestations with sensory-motor polyneuropathy and cardiac involvement.

According to the families studied, it is possible that there is an approximation of consanguinity between the first and third cases. However, this information cannot be validated by those who would present said consanguinity due to their death.

In case 1, his relatives mention having ancestors from Spain who arrived in Ecuador and traveled to Colombia and Peru, finally deciding to settle in Ecuador in the Province of Loja; in case 2, they mention having ancestors from Spain; and in case 3, they do not know their origins.

The present cascade study, which consisted of carrying out the genetic study on family members, allowed us to discover a total of 14 positive patients in the period from January 2020 to January 2024, constituting an effective method for carrying out research on relatives of I, II and III degrees of consanguinity (Figure 8).

Tests were carried out on a total of 44 people from these family trees, of which 14 are carriers of the gene with the p.Ser43Asn variant and are of an average age of 39 years (since their diagnosis through genetic study), with a history of origins in the Loja canton since its inception, and with current residence in the province and city of Loja, the province of Zamora and in the city of Guayaquil. Chronology Figure 1

The observation of the first cases allowed us to identify more families different from those mentioned in the article. Currently, as of January 2024, from the first families studied, there are a total of 22 cases carrying the variant coming from the same sector.

It is important to highlight that the increase in patients detected as carriers occurs after the cascade study was carried out: the genetic analysis of biological relatives of a patient with the hereditary pathogenic variant; this includes asymptomatic individuals without manifestations of the disease. The families found share an origin and, in some cases, residence in the city of Loja and the province of Loja, specifically the canton of Quilanga, as well as a history of death of family members with heart disease.

The so-called “founder effect occurs when a population is established from a small number of individuals extracted from an ancestral population” [10].

It is known that the phenotypic expression of hTTR amyloidosis is variable according to the type of mutation. Though its geographical origin may not be a predictor of the expression of the disease, it provides information about the associated variant [11,12].

Current pharmacological therapies seek to attack different points of amyloidogenesis: production, deposition and elimination.

There are so-called genetic RNA silencers (siRNAs) that cause selective degradation of the TTR mRNA, among these are: patisiran, revusiran and vutrisiran.

Among the antisense oligonucleotides (ASOs) are inotersen and eplontersen, which reduce the expression of TTR by introducing nonsense mutations in the TTR gene.

ASOs and siRNAs are designed to alter the disease phenotype of ATTR amyloidosis by degrading both wild-type and mutant TTR RNA transcripts, reducing the synthesis of TTR protein [13]. The two main gene silencing compounds that have undergone extensive clinical assessment in ATTR include inotersen (ASO) and patisiran (siRNA), and they have been approved for treating stage 1 and 2 ATTRv [14].

Patisiran (siRNA), which shows a reduction [15] of 85–96% after the second dose in adverse reactions related to infusion, and inotersen were shown through phase III clinical trials to improve the course of polyneuropathy in stages I and II as well as the quality of life of patients, as estimated by results on the score on the Norfolk Quality questionnaire reported by the patient [16].

As per the Life-Diabetic Neuropathy (QOL-DN) questionnaire, the adverse events found in a study of 117 patients were glomerulonephritis in 3% and thrombocytopenia in 2% as well as important adverse effects that require safety monitoring for early detection and treatment [16].

Among the main transthyretin stabilizing drugs are tafamidis and diflunisal. Tafamidis, or 2-(3,5-dichloro-phenyl)-benzoxazole-6-carboxylic acid, acts selectively by binding to TTR and stabilizing both wild-type TTR and mutant TTR. There are several clinical trials and extension studies that demonstrate its good efficacy and adequate tolerability. During the first double-blind placebo-controlled trial (Fx-005), four adverse events were identified: diarrhea, urinary tract infection, abdominal pain and vaginal infection [17]. These events were related to disease progression in subsequent studies.

Diflunisal, a non-steroidal anti-inflammatory drug, has also been shown to stabilize transthyretin; however, due to the presence of toxicity in patients with cardiomyopathy, it is not recommended [18].

Removing amyloid deposits is another important mechanism that is being investigated. The drugs considered to act within cardiac clearance are doxycycline and tauroursodeoxycholic bile acid (TUDCA), both with an acceptable safety profile [19]. EGCG (epigallocatechin-3 gallate)—the most abundant polyphenol in green tea, has shown a 6% to 12.5% reduction in ventricular mass [20,21].

Liver transplantation (LT) was the first approved treatment, acting by eliminating the production of amyloidogenic mutated TTR produced by the liver [22].

However, although effective, there are some limitations: it is a major surgical procedure inevitably associated with its inherent risks; there are a limited number of compatible grafts; and some patients still have disease progression due to continued formation of TTR amyloid fibrils [22].

In a systematic review on the results of isolated heart transplantation, a total of 123 patients were analyzed, finding favorable results with a mean survival of 4.33 years [23].

With current advances, a multidisciplinary approach, new therapies and adequate patient selection, there is an encouraging future.

In Ecuador, pharmacological therapies are not yet available, and there are certain access limitations to the availability of certain imaging studies (cardiac scintigraphy and cardiac magnetic resonance) and genetic tests that generate delays in the diagnosis of this disease.

It is important to know the possible epigenetic factors that establish the relationship between genetic determinants and external environmental factors, such as: diet, toxins, environmental polluting factors, physical activity and mental state that allow for the phenotypic expression of this variant [23].

The real incidence of amyloidosis in Ecuador and the existing variants are unknown, so continuing with these types of studies will allow us to identify the real data and to implement programs that will improve care and provide support for what is necessary for its diagnosis and treatment.

This variant is currently considered the most frequent variant, and it has a significant impact on the quality of life and a high mortality when there is cardiac involvement. The genetic determination of the first case and the identification of this variant allowed us to know that it existed in Ecuador

## Figures and Tables

**Figure 1 medsci-12-00058-f001:**
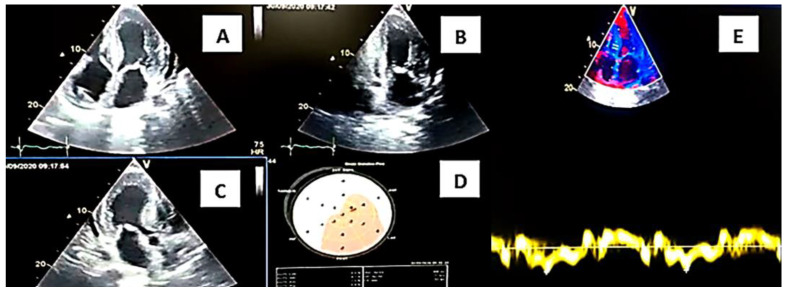
Transthoracic Doppler Echocardiogram Study. Source: authors, 2021. (**A**) 4-chamber apical section demonstrating septal thickening, interatrial septum and atrioventricular leaflets. (**B**) 2-chamber apical section demonstrating thickening of inferior and anterior walls. (**C**) 5-chamber apical section denoting ventricular thickening and mitral valve leaflets and visualization of the aortic valve. (**D**) Polar map with a total longitudinal global strain: −5%. Doppler ultrasound with (**E**) significant drop in tissue velocities is observed; e’ wave of 0.03 m/s at the septal level.

**Figure 2 medsci-12-00058-f002:**
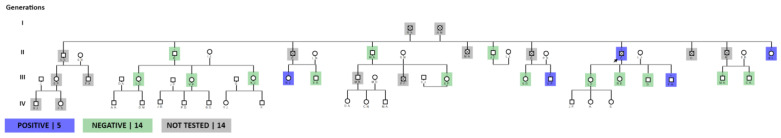
Source: authors, 2021. Family tree analyses.

**Figure 3 medsci-12-00058-f003:**
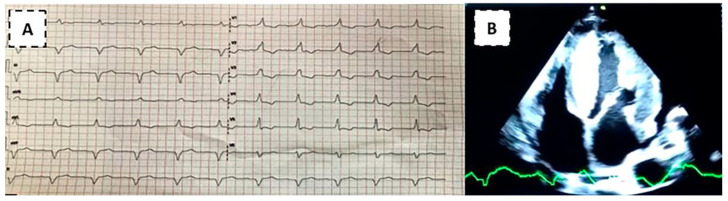
Source: authors, 2021. (**A**) Electrocardiogram demonstrating first degree AV block, complete right bundle branch block and inferior pseudo infarction. (**B**) Apical 4-chamber image shows a very thickened 24 mm interventricular septum and thickened interatrial septum, as well as the mitral valve leaflets and mild pericardial effusion.

**Figure 4 medsci-12-00058-f004:**
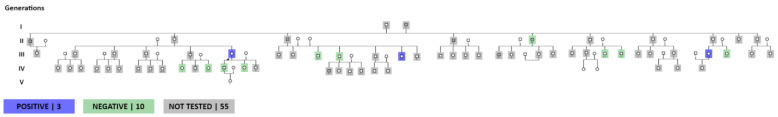
Source: authors, 2021. Family tree analyses.

**Figure 5 medsci-12-00058-f005:**
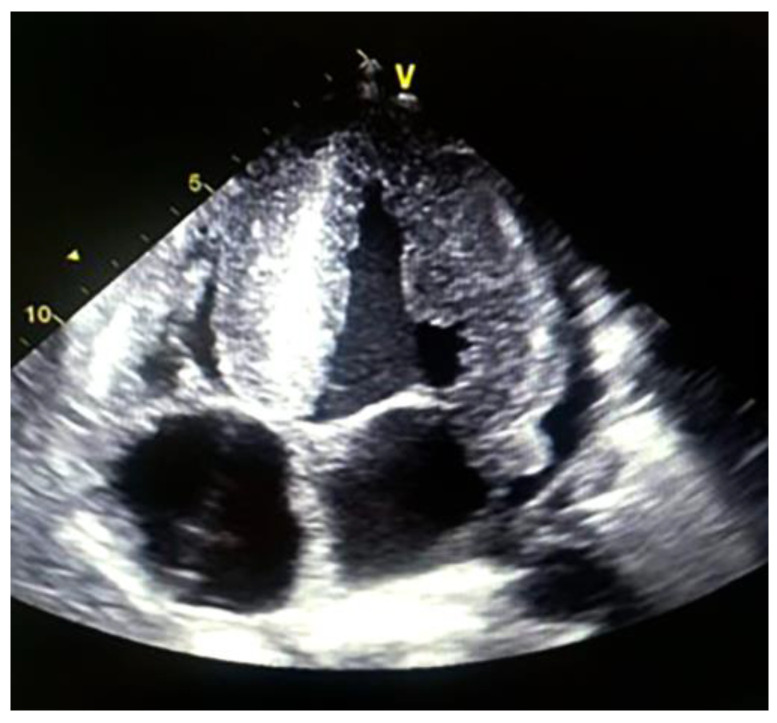
Source: authors, 2021. Apical 4-chamber image shows a very thickened 26 mm interventricular septum, thickened interatrial septum and atrioventricular leaflets and pericardial effusion.

**Figure 6 medsci-12-00058-f006:**
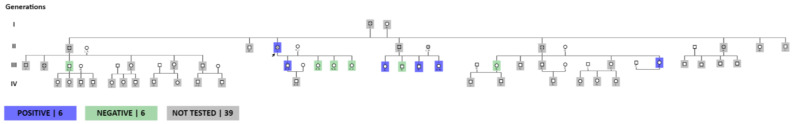
Source: authors, 2021. Family tree analyses.

**Figure 7 medsci-12-00058-f007:**
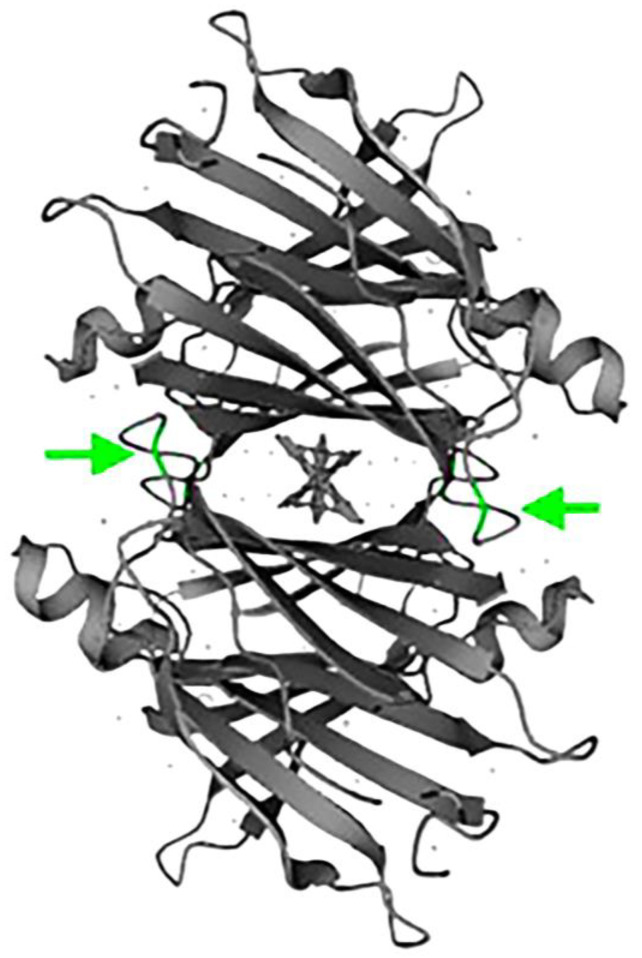
Amino acid positions (highlighted in green) subjected to the mutations described in Table 2. Source: biorender 2024s.

**Figure 8 medsci-12-00058-f008:**
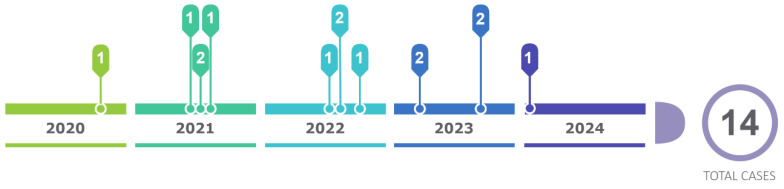
Chronological order.

**Table 1 medsci-12-00058-t001:** Demographic Characteristics. and Clinical.

**Demographic Characteristics**			
	Case 1	Case 2	Case 3
Age	59	49	62
Sex	Male	Male	Male
Place of origin	Quilanga	Loja	Quilanga
Place of residence	Loja	Loja	Quilanga
Parents’ place of origin	Quilanga	Quilanga	Quilanga
Occupation	Administrator	Businessman	Farmer
**Clinical and Imaging Characteristics**			
ECG findings	Inferior pseudo infarction, ventricular extrasystoles	Bifascicular block: RBBB, BAV first degree. Inferior pseudoinfarction	Complete AV block
Nyha functional class	III	II	III
Myocardiopathy on echocardiogram	Yes	Yes	Yes
Neurological evaluation/polyneuropathy neuropathy	Yes	No	Yes
Carpal tunnel syndrome	No	No	No
Gastrointestinal manifestations	Yes	Yes	Yes
Device placement	Yes (CDI)	No	No
Outcomes	Syncope-TBI/death	Referred to heart transplant	Cardiogenic shock/death

Source: authors, 2021s.

**Table 2 medsci-12-00058-t002:** Result of the analysis of the TTR Gene and its identified variant.

Name (Protein Variant incl. 20-aa Signal Peptide)	Sequence Variant (mRNA)	Genomic Location:	Codon Change	Location	Variant
Ser23Asn (p.Ser43Asn)	c.128G > A	chr18-31592954 G > A	AGT > AAT	Exon 2	missense

Source: Mutations in hereditary amyloidosis, Dorota Rowczenio & Ashutosh Wechelaker, 2024.

## Data Availability

The raw data supporting the conclusions of this article will be made available by the authors, without undue reservation.

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
