# Peer review of "Experience of Hereditary Amyloidosis with Rare Variant in Ecuador: Case Reports"

_medsci, 2024, doi:10.3390/medsci12040058_

Round 1
Reviewer 1 Report (New Reviewer)
Comments and Suggestions for Authors
This was a very interesting and impactful manuscript whose objective was to carry out a descriptive cross-sectional analysis on the presentation of transthyretin amyloidosis in families carrying the p.Ser43Asn mutation in the TTR gene. Overall it had much excellent information but needs some cleaning up and additional details. However, it remains very clinically impactful and of great importance to any clinician/geneticist.
Overall the paper was well-written but needs to be edited for the quality of the language.
As one example, “that causes a great impact on quality of life” should be “greatly impacts life”.
In the abstract please mention clearly the gene name (TTR) and the fact that p.Ser43Asn is the mutation you chose to focus your study on.
This applies throughout the whole manuscript. Gene vs variant vs mutation vs allele should be used consistently.
Here the manuscript is focused on the TTR p.Ser43Asn mutation, not the p.Ser43Asn gene. Please correct throughout.
In Table 1, consistent capitalization might make the table cleaner and more legible (of its contents, whether headers or details).
Even if it’s the same for each patient, could you include in Table 1 that patients’ mutation? This will offer a clean and quick summary of your papers’ clinical/genetic reportings.
Might you be able, in addition, to include a small recap table with the mutation in terms of its chromosome location (chr:…), coding sequence (c….) and protein localization (p.Ser43Asn).
This would be hugely helpful to any geneticist reading through the paper.
Might it be possible to clarify why you chose this mutation other than the fact that it was presented in your clinic? Is there any other reason (for example might this mutation be more frequent than others or more impactful than others in TTR)?
(You mention “this designation is due to a change in nomenclature”—just to be sure, this isn’t reflective of differentially spliced protein variants (e.g. canonical vs other forms)? This can be checked on UniProt or equivalent, but isn’t necessary at all—just a detail.)
(Also a non-necessary detail, you mention “the final protein structure is expressed from amino acids 21 to 147”—can you mention which functional parts of the protein this includes? This would be of interest to perform functional studies, and/or help support any impact predictions of the mutation.)
Can you use in silico prediction tools to predict how well conserved the affected region of the gene is (e.g. using its PhyloP score), or—at the very least (necessary)—the impact of the mutation (e.g. using its SIFT or PolyPhen score)? (The therapies you mention aim to decrease TTR levels so it would be interesting to understand/at least start to investigate the functional impacts on TTR of the mutation using basic gold standard tools.)
When you mention “In our city of Loja, capital of the province of Loja in the South of the country, our first case of Transthyretin amyloidosis was diagnosed in 2020 with genetic determination of the variant”, could you include here and thereafter the actual TTR variants of the cases you describe? Otherwise, readers can always reach out to know—but one such brief meta-analytic report would be hugely interesting.
Comments on the Quality of English LanguageSee above.
Author Response
Primer comentario
En el resumen, mencione claramente el nombre del gen (TTR) y el hecho de que p.Ser43Asn es la mutación en la que eligió centrar su estudio.
Respuesta
La determinación genética del primer caso nos ayudó a conocer que en Ecuador existía esta variante, que actualmente se considera la más frecuente, con importante impacto en la calidad de vida y alta mortalidad cuando hay compromiso cardíaco.
Descrito en el último párrafo de la discusión.
Segundo comentario
Esto se aplica a todo el manuscrito. Los términos gen, variante, mutación y alelo deben usarse de manera coherente.
En este caso, el manuscrito se centra en la mutación p.Ser43Asn de TTR, no en el gen p.Ser43Asn. Corrija todo.
Respuesta
Se revisó nuevamente, se corrigió y se aclaró en el primer párrafo de la discusión.
Tercer comentario
En la Tabla 1, el uso uniforme de mayúsculas podría hacer que la tabla (de su contenido, ya sean encabezados o detalles) sea más clara y legible.
Incluso si es la misma para cada paciente, ¿podría incluir en la Tabla 1 la mutación de ese paciente? Esto le permitirá obtener un resumen claro y rápido de los informes clínicos/genéticos de sus artículos.
Respuesta
Se corrigió la fuente de la Tabla 1 y se agregó la Tabla 2 con los datos solicitados.
|
Nombre (variante de proteína que incluye péptido señal de 20 aa) |
Variante de secuencia (ARNm) |
Ubicación genómica: |
Cambio de codón |
Ubicación |
Variante |
|
Ser23Asn (pág.Ser43Asn) |
c.128G>A |
chr18-31592954 G>A |
AGT > AAT |
Exón 2 |
sin sentido |
Cuarto comentario
¿Podrías aclarar por qué elegiste esta mutación además del hecho de que estaba presente en tu clínica? ¿Hay alguna otra razón (por ejemplo, podría ser esta mutación más frecuente que otras o tener más impacto que otras en el TTR)?
Respuesta
Sí, es la variante más común hasta ahora.
Su informe se produjo en realidad después de este primer caso reportado con resultados genéticos en 2020, del que surgen los demás.
Se menciona en el último párrafo de la discusión.
Cinco comentarios
(También es un detalle innecesario: mencionas que “la estructura final de la proteína se expresa desde los aminoácidos 21 al 147”; ¿puedes mencionar qué partes funcionales de la proteína incluye esto? Esto sería de interés para realizar estudios funcionales y/o ayudar a respaldar cualquier predicción del impacto de la mutación).
¿Puede utilizar herramientas de predicción in silico para predecir qué tan bien conservada está la región afectada del gen (por ejemplo, utilizando su puntuación PhyloP) o, como mínimo (necesario), el impacto de la mutación (por ejemplo, utilizando su puntuación SIFT o PolyPhen)? (Las terapias que menciona están destinadas a disminuir los niveles de TTR, por lo que sería interesante comprender/al menos comenzar a investigar los impactos funcionales de la mutación en TTR utilizando herramientas de referencia básicas).
La serina del codón 43 está moderadamente conservada y los análisis computacionales (SIFT: tolerada, PolyPhen-2: posiblemente dañina) predicen efectos contradictorios de esta variante en la estructura y función de la proteína. Según la información disponible, se considera que esta variante es probablemente patógena.
Respuesta
Se agregó un párrafo a la discusión con la información solicitada, se agregó un gráfico sobre la proteína y la ubicación del cambio resultante en verde, detallado en la discusión.
"Se describe en la literatura en individuos afectados por amiloidosis relacionada con transtiretina y se informa en ClinVar, Variation ID: 661615, (archivo público para respaldar la evidencia acumulada de la importancia clínica de las variantes genéticas), y no se encuentra en bases de datos de población general (Exome Variant Server, Genome Aggregation Database), lo que indica que no es un polimorfismo común.
Los análisis funcionales demuestran una estabilidad reducida del monómero y una formación positiva de amiloide en biopsias endomiocárdicas de múltiples trasplantes. La serina en el codón 43 está moderadamente conservada y los análisis computacionales (SIFT: tolerada, PolyPhen-2: posiblemente dañina) predicen efectos contradictorios de esta variante en la estructura/función de la proteína. Con base en la información disponible, se considera que esta variante es probablemente patógena.

Reviewer 2 Report (New Reviewer)
Comments and Suggestions for Authors
By this manuscript the Authors provide a description of 3 clinical cases of people affected by transthyretin amyloidosis with the pSer43Asn variant.
Case reports are interesting but limited to 3 cases in the range 2020-2022 whereas authors indicated that there are a total of 22 cases carrying the variant until January 2024; the addition of at least 3 more cases should add additional value to the cross-sectional analysis
As this manuscript is a case report, there Is no research activity behind hence it is a very descriptive report of clinical history of 3 patients. There is no additional value to the research field To answer to your points: • What is the main question addressed by the research? Manuscript is not a research article in which a specific question needs to be addressed but it is a case report with no questions to be addressed but in which 3 real cases ins a specific country are described. An extra-feature is the genetic segmentation/correlation for all the patients. • Do you consider the topic original or relevant to the field? Does it address a specific gap in the field? Please also explain why this is/ is not the case. As a case report for the mutation reported, manuscript is not adding additional value to the field because it is describing 3 case in the range 2020-2022. At the end of the paper authors indicated that much more cases have been registered till 2024 so I think an additional value would be an overall case overview rather than only 3 cases • What does it add to the subject area compared with other published material? No significant additional value compared to other similar manuscript • What specific improvements should the authors consider regarding the methodology? What further controls should be considered? This a case report hence a descriptive paper describing clinical history of the patients hence no mythology or controls have to be considered • Are the conclusions consistent with the evidence and arguments presented and do they address the main question posed? Please also explain why this is/is not the case. As a case report there are no results to be reportedAuthor Response
Pregunta 1.
“Los informes de casos son interesantes, pero se limitan a 3 casos en el rango de 2020 a 2022, mientras que los autores indicaron que hay un total de 22 casos portadores de la variante hasta enero de 2024; la adición de al menos 3 casos más debería agregar valor adicional al análisis transversal”.
Respuesta:
Se consideró en ese momento, porque no se habían encontrado otros casos similares en la institución hospitalaria, con cuadro clínico completo de amiloidosis y con confirmación del estudio genético.
El resto de los casos encontrados son posteriores al seguimiento del resto de los familiares de cada paciente índice, quienes no necesariamente presentan signos clínicos ni manifestaciones de la enfermedad en los estudios disponibles. Cabe destacar que en nuestra institución y ciudad no se cuenta con pruebas de imagen como resonancia magnética y/o estudios gramaticales cardíacos. Además, no existe una rápida realización y obtención de resultados de los estudios genéticos con la prontitud deseada.
Pregunta 2.
“¿Cuál es la pregunta que se plantea para la investigación?”
Respuesta:
¿Cómo se manifiesta la lamiloidosis por transtiretina en su variante p.Ser43Asn?
La difusión del conocimiento de una enfermedad rara, poco común, sigue siendo importante cuando su aporte se orienta a cambios de comportamiento en los procesos de atención en salud, especialmente a nivel local. Otra razón es que en la provincia de Loja se reporta el mayor número de portadores del gen p.Ser43Asn. La determinación genética del primer caso ayudó a conocer que en Ecuador existía esta variante (resultado de la prueba genética el 29 de octubre de 2020) que hasta ahora se considera la más frecuente, y tiene su impacto por la presentación de alta mortalidad cuando es de fenotipo cardíaco.
Pregunta 3
¿Consideras que el tema es original o relevante en el campo?
Respuesta
Sí, considero que es relevante y original, porque la difusión del conocimiento y la investigación, por más sencilla que sea su iniciación y con todas las dificultades, motiva a otras personas a realizar esfuerzos para aprender, a generar cambios en las políticas de salud, la necesidad de obtener un registro nacional de datos, y otras brechas que existen en el sistema de salud.
Existe este “efecto fundador”, considerado en nuestro trabajo como el cantón “Quilanga”, de donde provienen todas las familias, que presentan esta misma variante. Y al cual hemos acudido para realizar el seguimiento y creación de estos árboles genealógicos.
Dentro de las manifestaciones clínicas iniciales, como las molestias gastrointestinales, se presentó como primera manifestación de la enfermedad confundiéndose con otros diagnósticos (síndrome del intestino irritable) en el primer y tercer caso. No se encontró en el segundo caso, lo que habla a favor de una variante de fenotipo mixto. Conocer qué factores influyen en su expresión son otras de las preguntas que surgen de la lectura del artículo.
Pregunta 4 y 5
¿Las conclusiones son coherentes con la evidencia y los argumentos presentados y abordan la cuestión principal planteada? Explique también por qué es así o no. Como se trata de un informe de caso, no hay resultados que comunicar.
Respuesta
Se consideró clasificarlo como “serie de casos” en base a que la variante mencionada comparte una característica común. Existieron limitaciones para incluir más casos por no cumplir con el estudio genético y otras pruebas diagnósticas, al ser un estudio estrictamente descriptivo.
Considerando sus contribuciones en la revisión, hemos decidido cambiar la mención de “serie de casos” por informe de caso.

This manuscript is a resubmission of an earlier submission. The following is a list of the peer review reports and author responses from that submission.
Round 1
Reviewer 1 Report
Comments and Suggestions for Authors
In the manuscript, Luzuriaga Carpio et al shared cross-sectional descriptive analysis on the appearance of hereditary Transthyretin amyloidosis (ATTR) in families carrying the rare p.Ser43Asn variant on the index case identified in the period from Jan 2020 to Jan 2022 at Hospital General Manuel Ygnacio Monteros-IESS, Loja, Ecuador. Transthyretin amyloidosis (ATTR) is incited when a transthyretin (TTR) protein, which is normally produced by the liver, pathologically misfolds, aggregates into amyloid fibrils, and deposits in various tissues causing irreversible damage. Transthyretin amyloid cardiomyopathy (ATTR-CM) is an increasingly diagnosed condition causing heart failure. Although wild-type transthyretin amyloidosis (ATTRwt) is the most frequent form of ATTR-CM, hereditary transthyretin amyloidosis (ATTRv) can also occur. It has been proposed that early and prompt clinical diagnosis is the holy grail for the effective therapy of ATTR-CM, especially for genotypes associated with a rapid progression (such as p.Ser43Asn). In the present manuscript authors used epidemiology, consanguinity, Cardiac magnetic resonance, echocardiography, and ECG to assess the disease manifestations. However, they excluded the 99mTc‐labelled bone radiotracer scintigraphy data considered critical for the diagnosis. Also, the study is done with very small number of patient population and there is no validation of the data to rule out false positives and negatives. The work lacks rigor and significance to provide clinical guideline for the diagnosis of ATTR-CM and specific disease-modifying treatment, leading to improved functional outcomes and prolonged survival. The narrative compiles the clinical experience of only three index cases without critical evaluation of subject matter in an informative and insightful fashion and thus not recommended for publication in Medical Sciences.
Comments on the Quality of English LanguageFor example Under Discussion, Line 212-214:
"According to the work published by Maria Papahanasiou and colb, published in December 2020, it documents a report of 2 opportunely diagnosed patients from a family in Italy whose identified variant is the same as our cases and which is considered to have a clinically aggressive course. due to its exclusively cardiac phenotype."
The sentence can be simplified as:
"Maria Papahanasiou et al reported 2 opportunely diagnosed patients from a family in Italy with genetic variant similar to the one identified in our cases and which is considered to have a clinically aggressive course, due to its exclusively cardiac phenotype."
Author Response
Saludos estimado revisor, hemos respondido cada punto del siguiente manuscrito y hemos subrayado las correcciones en amarillo.
Queremos hacer de su conocimiento que nuestro trabajo son reportes de casos clínicos poco conocidos en nuestro medio en los cuales hemos realizado un arduo trabajo de exploración y recolección de datos. En nuestro medio existen muchas limitaciones debido a que pocas personas conocen sobre esta enfermedad. Los invito a reconsiderar el manuscrito ya corregido que adjunto en word.
Concluimos que nuestro trabajo no es una guía clínica sino un reporte de caso clínico de una enfermedad como la amiloidosis. Un cordial saludo.

Reviewer 2 Report
Comments and Suggestions for Authors
Transthyretin amyloidosis, a severe and underdiagnosed multisystem disorder in Ecuador, has a profound impact on quality of life and increases mortality rates, especially when it involves the heart. Addressing the significant healthcare gaps in Ecuador, this study underscores the critical need for timely diagnosis of transthyretin amyloidosis. The paper presents a cross-sectional descriptive analysis focusing on the manifestation of the disease in families carrying the p.Ser43Asn on TTR gene. (1) The first case reported is A 59-year-old male from Loja who passed away in early 2021. Genetic testing later confirmed he was heterozygous for the p.Ser43Asn variant associated with amyloidosis. A retrospective analysis revealed a prolonged misdiagnosis, tracing back to the initial symptoms reported in 2013. Family tree analysis indicated that, of 19 family members tested, 5 tested positive for the amyloidosis-associated variant. The family predominantly resides in the Province of Loja. (2) The second patient, a 49-year-old from Loja, also carries the p.Ser43Asn variant with confirmed heterozygosity. Family tree analysis reveals that 3 out of 13 members tested positive for the variant. Additionally, 4 individuals from the second generation are died of myocardial infarction and heart failure. This family group has members residing in the Provinces of Loja, Zamora, and Guayas, as well as in Spain. (3) The third case involves a 62-year-old male from the province of Loja, who was diagnosed with third-degree AV block, hypertrophic cardiomyopathy, and heart failure. He carried the p.Ser43Asn variant with heterozygosity and passed away in 2023. Family tree analysis indicates that 7 out of 8 siblings died with symptoms of generalized weakness, cachexia, and dyspnea, leading to sudden death, without an established diagnosis.
Comments:
The case report is logically presented and concisely described. However, the discussion section from lines 233 to 267 appears overly dense and requires revision for better clarity and flow.
Specific comments and suggestions:
1. Indicate p.Ser43Asn variant is at TTR gene when it first appears in the paper.
2. Line 72 “2 years. months”
3. Line 118, “test. test”
4. Line 150, 153 and 185, font size and format.
5. Figure 5 and 6 legend format.
6. Line 215 “course. due”.
7. Line 245 “polyneuropathy. as”.
Additional comments:
This paper investigates the genetic variance of amyloidosis in three Ecuadorian families, focusing specifically on the TTR p.Ser43Asn variant of transthyretin amyloidosis. It examines the variant's penetrance, associated phenotypes, and potential regional concentration, suggesting endemic occurrence in Ecuador. The study fills a significant gap in understanding amyloidosis within the Ecuadorian population, highlighting its impact on affected families and exploring genetic, clinical, and regional aspects.
While the focus on a specific genetic variant within a defined population addresses an important gap, the methodology section could benefit from a more detailed description of the genetic testing processes and selection criteria. This would enhance the replicability and robustness of the findings.
The conclusions are appropriately cautious but could be further strengthened by discussing the study's limitations, such as its potential lack of generalizability to other populations. Additionally, the implications for genetic counseling and public health policy in Ecuador could be explored more thoroughly to provide actionable insights for healthcare providers and policymakers.
The references provide a solid context for the study, but a deeper comparison with similar studies from other regions or different methodologies could broaden the understanding of transthyretin amyloidosis's global implications.
Overall, this manuscript offers valuable insights into the regional characteristics of transthyretin amyloidosis in Ecuador. By addressing the suggested improvements, it can make a significant contribution to the field, potentially leading to targeted public health interventions and enhanced diagnostic protocols within the region
Comments on the Quality of English LanguageComments:
The case report is logically presented and concisely described. However, the discussion section from lines 233 to 267 appears overly dense and requires revision for better clarity and flow.
Specific suggestions:
1. Indicate p.Ser43Asn variant is at TTR gene when it first appears in the paper.
2. Line 72 “2 years. months”
3. Line 118, “test. test”
4. Line 150, 153 and 185, font size and format.
5. Figure 5 and 6 legend format.
6. Line 215 “course. due”.
7. Line 245 “polyneuropathy. as”.
Author Response
In our country there are many limitations for the diagnosis of this disease as amyloidosis, but all the authors have carried out examinations in this work and follow-up in order that the international community knows about this pathology catalogued as a rare disease, so we have reported in this work three clinical cases with the respective follow-ups and documentation.
Attached is the word file with the corrections that you have suggested.
Kind regards
